# The Dose–Response Relationships of Different Dimensions of Physical Activity with Daily Physical Function and Cognitive Function in Chinese Adults with Hypertension: A Cross-Sectional Study

**DOI:** 10.3390/ijerph182312698

**Published:** 2021-12-02

**Authors:** Meng Ding, Ningxin Jia, Yanan Zhou, Bin Li

**Affiliations:** College of Physical Education, Shandong Normal University, Jinan 250014, China; 2020020238@stu.sdnu.edu.cn (N.J.); 2020020240@stu.sdnu.edu.cn (Y.Z.); lb001600@126.com (B.L.)

**Keywords:** hypertension, physical activities, daily physical function, cognitive function

## Abstract

Objective: The purpose of this study was to explore the dose–response relationships of different dimensions of physical activity (intensity, time, frequency, capacity, and metabolic equivalent) with daily physical function (DPF) and cognitive function (CF) in Chinese adults with hypertension. Methods: The 6216 hypertensive patients included in this study were from the China Health and Retirement Longitudinal Study (CHARLS), which was conducted in 2015. Physical activity (PA) was divided into vigorous PA (VPA), moderate PA (MPA), and light PA (LPA). Linear regression models and binary logistic regression models were established to assess the associations with indicators. Results: Patients with VPA have a lower probability of impaired DPF; however, patients with VPA had lower CF scores. Patients with nearly all the aspects of MPA have lower rates of impaired DPF and higher CF scores (*p* < 0.05). Patients with LPA have a lower probability of impaired DPF and higher CF scores. In addition, patients with between 1800 and 2999 MET-minutes per week had the lowest rates of impaired DPF (OR = 0.10, 95% CI 0.02, 0.39) and the highest CF scores (β = 3.28, 95% CI 2.25, 4.31). Conclusions: This study found that patients with hypertension with moderate-intensity physical activity (nearly all aspects) and LPA had better DPF and CF. The best daily physical function and CF was with METs of 1800–2999 min per week. However, VPA should be recommended with caution in Chinese adults with hypertension.

## 1. Introduction

Hypertension has been a major health challenge nationally and even worldwide [1,2]. As of 2015, there have been more than 1.1 billion people suffering from hypertension in the world, and China accounts for roughly one-fifth of them [3]. In addition, the prevalence of hypertension is still on the rise and will likely continue to rise in the future [4]. In the literature, studies have reported that hypertension and the complications thereof negatively impact the physical and mental health of patients, leading to a heavy economic burden on families and on society, as well as consuming a large amount of medical and social resources [5].

Recent studies have shown that the increased degree of arterial stiffness in patients with hypertension leads to a decrease in the level of brain-derived neurotrophic factors in the hippocampus, which may cause memory impairment, cognitive impairments such as vascular dementia [6,7], as well as negative changes in the volume and function of the prefrontal lobe cortex and its lower regions. As a result, executive functions such as control, attention, working memory, and judgment may be impaired [8]. In addition, long-term treatment of hypertension affects patients’ daily activity abilities [9,10], which is usually manifested as dysfunction of motility, cognition, and emotional expression [11]. All these are attributed to the side effects of antihypertensive drugs [12]. However, hypertension, as a lifelong disease, has no radical treatment at present, and patients need to take drugs for a long time to control their blood pressure [13,14]. It is worth noting that the latest guidelines encourage hypertensive patients to engage in moderate physical activity (PA) on a regular basis [15], and related studies have pointed out that improving and controlling hypertension through lifestyle measures such as physical activity can result in significant social, public health, and economic benefits [16].

Although physical activity has attracted wide attention for improving the physical and mental health of patients with hypertension, to the best of our knowledge, currently, there is a lack of large-sample studies for verification, and no study has explored the different dimensions of physical activity and the dose–response relationship of metabolic equivalent (MET) with daily physical function (DPF) and cognitive function (CF) in patients with hypertension in China. Therefore, the purpose of this study is to explore the dose–response relationships of different dimensions of physical activity including intensity, time, frequency, capacity, and metabolic equivalent with DPF and CF in Chinese adults with hypertension. We hypothesized that there was a dose–response relationship between different dimensions of physical activity (including intensity, time, frequency, volume and metabolic equivalent) and DPF and CF in Chinese adult patients with hypertension.

## 2. Materials and Methods

### 2.1. Participants and Study Design

The data of this study came from the China Health and Retirement Longitudinal Study (CHARLS), which was conducted in 2015. CHARLS is a nationally representative longitudinal cohort study with a baseline survey conducted from 2011 to 2012 and followed up every two years among people aged 45 and above in China. To estimate social, economic, and health conditions in China, the study used a four-stage stratified cluster sampling method to select respondents from 450 villages in 150 county-level units across 28 provinces in China. This study was approved by the Biomedical Ethics Review Committee of Peking University (IRB00001052-11015). All participants provided written informed consent.

A total of 21,096 people took part in the 2015 study. Subjects had complete data including age, height, weight, gender, education level, marital status, smoking status, alcohol consumption status, physical activity records, hypertension diagnosis records, and so on. Samples with missing data, outliers, or logic errors were excluded. Finally, 6216 hypertensive patients were included in the final statistical analysis of this study.

### 2.2. Assessments

#### 2.2.1. Screening of Hypertensive Patients

Patients with hypertension were screened by responding to two questions: “Has a doctor ever told you that you have high blood pressure?” and “Are you currently treating hypertension and its complications in any of the following ways?” We believe that when the answer to one or both of the above two questions was “yes”, the patient had hypertension. For example, if a patient answered that they had been diagnosed with hypertension by a doctor, regardless of whether they had received treatment (such as taking medicine), the patient was considered to suffer from hypertension.

#### 2.2.2. Physical Activity

Each participant reported weekly PA, including vigorous physical activity (VPA) (i.e., PA that makes you feel short of breath, such as carrying heavy objects, digging, farming, aerobic exercise, rapid cycling, bicycle loading, etc.), moderate physical activity (MPA) (i.e., PA that makes you breathe faster than usual, such as carrying light goods, cycling at regular speed, mopping, Tai Chi, brisk walking, etc.), and light physical activity (LPA) (i.e., walking from one place to another at home or at work, and other walking activities for leisure, sports, exercise or entertainment). Each subject was asked, “Do you conduct VPA/MPA/LPA for at least 10 min continuously in a usual week?” If the response was “no”, they were considered to be taking no VPA/MPA/LPA. If the answer was “yes”, they were asked the following additional questions: “How many days do you normally take VPA/MPA/LPA in a week?” and “How long do you spend on VPA/MPA/LPA each time?” The frequency of PA ranged from 0–7 d/w and was separated into 4 levels: no activity (0 d/w), 1–2 d/w, 3–5 d/w, and 6–7 d/w. The duration of PA was categorized into 5 levels: no activity, 10–29 min/d, 30–119 min/d, 120–239 min/d, and ≥240 min/d. Considering that the average of each duration was used in the questionnaire instead of the specific duration [17], and following the WHO guidelines [18], the volume of VPA was divided into 4 levels: no activity, 10–74 min/w, 75–299 min/w, and ≥300 min/w. The volume of MPA/LPA was separated into 4 levels: no activity, 10–149 min/w, 150–299 min/w, and ≥300 min/w.

According to previous studies [19], 1 MET refers to the amount of oxygen consumed at rest, VPA can be expressed as 8 METs, MPA can be expressed as 4 METs, and LPA can be expressed as 3.3 METs. The total physical activity (i.e., METs) was calculated as the sum of scores for VPA + MPA + LPA [19]. According to the guidelines of the American College of Sports Medicine, the minimum level of total physical activity that is beneficial to health is defined as 600 MET-minutes/week [20]; therefore, MET-minutes/week was divided into 9 categories (0–599, 600–1199, 1200–1799, 1800–2999, 3000–5999, 6000–8999, 9000–11,999, ≥12,000).

#### 2.2.3. Daily Physical Function

DPF was measured by the Activities of Daily Living (ADL) and Instrumental Activities of Daily Living (IADL) scales [20]. Each option of both scales was divided into four levels: “no difficulties = 0”, “difficulties but still can be completed = 1”, “difficulties, need help = 2”, and “unable to complete = 3”. The range of composite scores was 0–43, and a composite ADL/IADL score of ≥11 was defined as “any functional loss”, and <11 was defined as “no functional loss”. Reliability of the questionnaire was analyzed with Cronbach’s alpha (α = 0.901).

#### 2.2.4. Cognitive Function

CF was evaluated through three types of tests: a telephone interview of cognitive status (TICS) (orientation and attention); word recall (episodic memory); and figure drawing (visuospatial ability) [21]. For the TICS (orientation and attention) [22] test, the subjects were asked to state the current year, month, date, day of week, and season; additionally, they were asked to count by subtracting seven from 100, and then keep subtracting seven from the answer until they heard stop. Then, the scores of the correct question were summed. For the word recall (episodic memory) [21,23,24] test, subjects were asked to remember and immediately recall as many words as possible, in arbitrary order, after reading a page consisting of 10 Chinese nouns (immediate recall); then, after 4 to 10 min, subjects were asked to recall as many original words as possible (delayed recall). The episodic memory score was the average score of the immediate recall and delayed recall. For the figure drawing (visuospatial ability) [21,23] test, subjects were shown a picture of two overlapping pentagons, and were asked to draw a similar image, with a score of 1 for success and 0 for failure. CF was scored as the sum of the scores of the three tests described above, with higher scores indicating better CF. Reliability of the questionnaire was analyzed with Cronbach’s alpha (α = 0.794).

#### 2.2.5. Covariables

The covariates considered in this study included age, sex (male or female), educational status (junior high school or below, senior high school or vocational school, college or above), marital status (married or partnered, separated, divorced or widowed, or never married), drinking (never, former, or current), and smoking (never, former, or current).

### 2.3. Data Analysis

In this study, a continuous variable was represented as a mean value and standard deviation (mean ± standard deviation), while a classified variable was represented as a number (*n*) and percentage (%). In addition, a linear regression or binary logistic regression model was used to analyze the different dimensions of physical activity and the dose–response relationships between metabolic equivalent and DPF and cognitive level in patients with hypertension. The linear regression data were expressed as β coefficient and 95% confidence interval (CI), while the binary logistic regression data were expressed as odds ratio (OR) and 95% confidence interval (CI). All models were adjusted for age, sex, educational status, marital status, alcohol consumption, smoking, and body mass index (BMI), potentially confounding covariates. *p* < 0.05 was statistically significant. SPSS software was used for all statistical analyses (Statistics 23, IBM Corporation, New York, NY, USA).

## 3. Results

The study included 6216 hypertensive patients, comprising 2910 males (46.8%) and 3306 females (53.2%). The mean age and BMI were 64.88 years and 25.26, respectively. Among the patients, the proportion who were married or living with a partner (82.9%), who had a junior high school education level or below (81.5%), who did not smoke (59.7%) and who did not drink alcohol (63.3) was even larger (Table 1).

The relationship between the frequency of physical activity and DPF (Table 2) showed that patients with high-intensity (6–7 days/week, OR = 0.33), moderate-intensity (1–2 days/week, OR = 0.24; 3–5 days/week, OR = 0.23; 6–7 days/week, OR = 0.26), and low-intensity (6–7 days/week, OR = 0.68) levels of physical activity had a lower probability of impaired physical function than patients with 0 days/week of physical activity. In addition, the relationship between frequency of physical activity and CF (Table 2) showed that patients with both moderate (1–2 days/week, β = 2.60; 3–5 days/week, β = 1.88; 6–7 days/week, β = 2.15) and low (3–5 days/week, β = 2.01; 6–7 days/week, β = 0.94) levels of physical activity had higher CF scores compared with those with 0 days/week of physical activity. However, patients with a high-intensity level of physical activity for 6–7 days a week (β = −0.92, 95% CI −1.66, −0.18) had lower CF scores.

The relationship between the duration of physical activity and DPF (Table 2) showed that patients with hypertension with moderate (10–29 min/day, OR = 0.31; 30–119 min/day, OR = 0.21; more than 240 min/day, OR = 0.12) and high intensity levels of physical activity for 240 min/day (OR = 0.31, 95% CI 0.13, 0.79) had a lower probability of impaired DPF compared with patients with 0 min/day of physical activity. The relationship between physical activity duration and CF is shown (Table 2). As compared with patients with 0 min/day of physical activity, patients with all durations of moderate physical activity (10–29 min/day, β = 2.49; 30–119 min/day, β = 2.57; 120–239 min/day, β = 1.52; more than 240 min/day, β = 1.87), and LPA (10–29 min/day, β = 1.45 and 30–119 min/day, β = 1.20) had higher CF scores. However, patients with high intensity physical activity for at least 240 min per day (β = −0.85, 95% CI −1.64, −0.05) had lower CF scores.

The relationship between physical activity capacity and DPF (Table 2) showed that patients with high (OR = 0.40, 95% CI 0.21, 0.78) and low intensity (OR = 0.66, 95% CI 0.49, 0.90) levels of physical activity for more than 300 min per week had a lower probability of impaired DPF compared with patients with 0 min per week. Patients with moderate-intensity physical activity from 10 to 149 min per week (OR = 0.21, 95% CI 0.08, 0.58) and for more than 300 min per week OR = 0.25, 95% CI 0.14, 0.44) had lower rates of impaired DPF. The relationship between physical activity capacity and CF is shown in Table 2. As compared with 0 min/week of physical activity, patients with more than 300 min/week of vigorous physical activity (β = −0.86, 95% CI −1.51, −0.21) had lower CF scores, and patients with moderate and LPA at any volume had higher CF scores (*p* < 0.05).

The relationships between metabolic equivalent (MET) values and DPF and CF (Table 3) showed that patients with at least 1200 MET-minutes/week had a lower probability of impaired DPF compared with inactive patients (*p* < 0.05). Moreover, patients with at least 600 MET-minutes per week had lower CF scores (*p* < 0.05). In addition, patients with between 1800 and 2999 MET-minutes per week had the lowest rates of impaired DPF and the highest CF scores.

## 4. Discussion

The hypertensive patients included in this study with a high frequency, high duration, and high volume of strenuous physical activity had a lower probability of impaired physical function, but poorer CF. Nearly all the aspects of moderate physical activity were associated with lower rates of impaired DPF and better CF. Hypertensive patients with LPA, high frequency and high volume had better daily physical function and CF. Patients with high blood pressure had the best daily physical function and CF with metabolic equivalents (METs) of 1800–2999 min per week.

Regarding intense exercise, related studies have shown that the systolic and diastolic blood pressure of patients after high-intensity exercise was lower than in patients after moderate intensity exercise [25], and it has been shown that high-intensity exercise has a positive effect on blood pressure control of patients with hypertension [26]. Previous studies have confirmed that high intensity exercise improved the muscle strength of old people, the ability of coordination, balance, and flexibility, etc., as well as improving DPF [27] for populations such as ours with an average age of 64.88 years. This is consistent with the conclusion of this study that high intensity exercise in hypertensive patients had a lower probability of impaired DPF. Additionally, related studies have shown that long-term moderate exercise could significantly improve the memory ability of patients with hypertension [28], while high intensity exercise did not change the accuracy of executive function and memory tasks in patients with hypertension [29]. However, high intensity exercise has been shown to be associated with a higher risk of cardiovascular complications and lower compliance, which was not conducive to the development of CF in hypertensive patients [30]. This may explain why, in this study, we found that hypertensive patients with high levels of exercise had worse CF, and that, compared with patients who did not engage in strenuous physical activity, those with high blood pressure and PA with a frequency of 6–7 days/week, duration of at least 240 min/day, and volume of at least 300 min/week had a lower rate of impaired DPF, but poorer CF. Therefore, high-intensity exercise should be recommended with caution for patients with hypertension.

In this study, hypertensive patients with moderate intensity physical activity at any frequency, duration, and volume level had lower rates of impaired daily physical activity and better CF. Previous studies have shown that moderate physical activity has more positive effects for improving memory and could fully improve the memory ability of hypertensive patients [31,32]. Moderate physical activity more positively influences overall cognitive function, as well as the ability to prevent neuronal degenerative diseases, which likely maintained CF [33]. In addition, studies have confirmed that moderate intensity aerobic exercise can improve adaptability of myocardium and skeletal muscle, and at the same time change the signal transduction of central angiotensin, and play the role of lowering blood pressure and heart rate, therefore, alleviating poor blood pressure [34,35]. In addition, hemorheology researchers have concluded that the cause of hypertension was possibly the change in hemorheology. It has been shown that moderate intensity exercise can significantly reduce the blood viscosity of patients with hypertension, thus, reducing peripheral resistance and improving hemorheology [36]. In addition, by enhancing the sensitivity of baroreceptors, reducing the secretion of norepinephrine, decreasing peripheral resistance, while also improving insulin sensitivity and changing the expression of vasodilator and systolic factors, blood pressure can be reduced [37] and DPF can be improved. Relevant guidelines also suggest that the exercise intensity of patients with hypertension should usually be moderate [16]. These studies support the findings of this study that moderate-intensity physical activity, in almost all aspects, is associated with a lower probability of impaired DPF, as well as better CF. In this study, it was concluded that hypertensive patients who undertook more than 300 min of moderate intensity exercised per week had better DPF and CF, which was consistent with previous studies suggesting that hypertensive patients could undertake more than 300 min of moderate intensity exercise per week [38]. Therefore, moderate-intensity exercise may be recommended as the best intensity for improving CF and the ability to do daily activities in patients with hypertension.

Patients with LPA of 6 to 7 days per week and volume of at least 300 min per week had lower rates of impaired physical function and better CF. Current studies have shown that low-intensity exercise may significantly improve the memory ability of patients with hypertension by increasing the effective perfusion of hippocampal blood volume [39]. In addition, long-term low-intensity exercise can improve working memory by improving hippocampal function in patients with hypertension; increase the proliferation of nerve cells; generate synapses and new blood vessels; increase the cerebral blood volume of hippocampal dentate gyrus; and promote the growth of cortex, cerebellum, striatum, and hippocampus [40]. Epinephrine is involved in emotional memory consolidation. Exercise causes the release of adrenaline, glucocorticoid, and cortisol, which play neuroprotective roles by increasing the secretion of brain-derived neurotrophic factor, improving memory ability, and preventing neurologic dementia caused by hypertension [41,42]. Moreover, long-term low-intensity exercise can also improve aortic elasticity in patients with hypertension and can have a more positive impact on the CF and cardiovascular health of patients [43]. Related studies have shown that exercise of 150 min or more per week in hypertensive patients can have positive effects on blood pressure control [44]. As a result, patients with hypertension who exercise 6 to 7 days per week and at least 300 min per week had better daily physical function and CF. This is consistent with the results of this study which showed that hypertensive patients with high frequency and high volume of low-intensity exercise had better daily physical function and CF. However, some studies have shown that, compared with moderate intensity exercise, low intensity exercise had less impact on CF [45]. We did not further divide low-intensity physical activity into more detailed physical activity intensity. Some studies have found that high-intensity physical activity had a significant correlation with health, while low-intensity physical activity had no correlation with health [46]. We suspect that this is one of the reasons why the duration of low-intensity physical activity in this study was not associated with any relationship between DPF and cognitive function.

Regarding the metabolic equivalent, previous studies have shown that neurogenesis is one of the closest and earliest links between exercise and cognition [47], and exercise could improve cognition by promoting adult hippocampal neurogenesis [48]. Some studies have also demonstrated that exercise could reduce overall cognitive decline and behavioral problems in patients with mild cognitive impairment or dementia by slowing the decline of working memory, and could also improve language ability [49]. In addition, exercise of at least 1200 METs has been shown to have a significant effect on overall cognition [34]. This is consistent with the results of this study, which showed that hypertensive patients with at least 1200 MET-minutes of physical activity per week had better daily physical function and CF. In addition, related studies have shown that exercise in patients with hypertension was helpful for improving the flexibility of the brain, and enabled patients to have a certain spatial imagination ability [36]. Furthermore, cardiorespiratory function or aerobic capacity was related to the improvement of CF [50]. Exercise intensity can improve the CF of the brain more than exercise duration; therefore, appropriate exercise intensity has a more significant effect on improving CF and delaying cognitive decline [51]. Exercise increases the oxygen supply to the brain by increasing vital capacity and VO2 max, and improves the levels of neurotransmitters such as acetylcholine, norepinephrine, dopamine, 5-hydroxytryptamine, and gamma-aminobutyric acid, as well as brain-derived neurotrophic factor, nerve growth factor, and neurotrophic factor. As such, the short-term memory ability, reaction speed, and anti-interference ability of older adult CF can be significantly improved [52]. It is worth noting that some studies have shown that a metabolic equivalent of 1800–2999 per week was more effective in improving cardiopulmonary endurance and improving vascular endothelial function and insulin sensitivity, as well as providing other health benefits [23]; this is consistent with the results of this study. Therefore, perhaps 1800–2999 MET-minutes/week is the optimal exercise intensity for improving daily physical function and CF in patients with hypertension.

This study is the first to investigate the relationships among different dimensions of physical activity, metabolic equivalent, DPF, and CF in patients with hypertension. In addition, this study used a large sample that was representative of China; therefore, these results can be generalized to the entire Chinese population. However, the limitations of this study should also be taken into consideration. First, this was a cross-sectional study, and as such, cannot accurately explain and analyze causality. In the future, longitudinal studies should be undertaken to further analyze causality. Secondly, the physical activity questionnaire we used was not a mature questionnaire, and the subjectivity of the questionnaire may have problems with insufficient measurement accuracy. Thirdly, we did not consider other diseases (such as early-onset Alzheimer’s disease and other dementias and degenerative diseases of the joints), which may have had an impact on our results. Therefore, considering the limitations of this study, high quality studies are still needed for further in-depth discussion.

## 5. Conclusions

This study found that patients with hypertension with moderate-intensity physical activity (nearly all aspects) and LPA had better DPF and CF. The best daily physical function and CF was with METs of 1800–2999 min per week. However, VPA should be recommended with caution in Chinese adults with hypertension. Rigorous randomized controlled trials or long-term cohort studies are still needed to provide stronger evidence in the future.

## Figures and Tables

**Table 1 ijerph-18-12698-t001:** Basic characteristics of participants.

Characteristic	Overall Sample (*n* = 6216)	
	Mean	SD
Age (year)	64.88	10.13
BMI (kg/m^2^)	25.26	3.69
	** *n* **	**%**
Sex		
Male	2910	46.8
Female	3306	53.2
Marital status		
Married or partnered	5153	82.9
Separated, divorced, or widowed	1016	16.3
Never married	47	0.8
Educational status		
Junior high school or below	5065	81.5
Senior high school or vocational school	992	16
College or above	159	2.6
Smoking		
Never	3713	59.7
Former	1016	16.3
Current	1487	23.9
Drinking		
Never	3935	63.3
Former	315	5.1
Current	1966	31.6

Abbreviation: SD, standard deviation; BMI, Body Mass Index. Data are presented as *n* (%) or mean ± SD.

**Table 2 ijerph-18-12698-t002:** Associations between PA and DPF and CF of patients with hypertension.

Variables	*n* (%)	Daily Physical Function		Cognitive Function	
		OR	95% CI	β	95% CI
Frequency					
VPA					
0 d/w	5436 (87.45)	1		1	
1–2 d/w	131 (2.11)	0.19	0.03, 1.37	−0.40	−1.69, 0.88
3–5 d/w	182 (2.93)	0.61	0.22, 1.73	−0.79	−1.91, 0.33
6–7 d/w	467 (7.51)	0.33	0.14, 0.76 **	−0.92	−1.66, −0.18 *
MPA					
0 d/w	4803 (77.27)	1		1	
1–2 d/w	204 (3.28)	0.24	0.07, 0.77 *	2.60	1.52, 3.67 **
3–5 d/w	273 (4.39)	0.23	0.08, 0.65 **	1.88	0.93, 2.83 **
6–7 d/w	936 (15.06)	0.26	0.14, 0.45 **	2.15	1.54, 2.76 **
LPA					
0 d/w	3998 (64.32)	1		1	
1–2 d/w	95 (1.53)	1.23	0.52, 2.93	1.34	−0.16, 2.84
3–5 d/w	237 (3.81)	1.04	0.57, 1.88	2.01	1.02, 2.99 **
6–7 d/w	1886 (30.34)	0.68	0.51, 0.91 **	0.94	0.45, 1.42 **
Duration					
VPA					
0 min/d	5425 (87.27)	1		1	
10–29 min/d	29 (0.47)	N/A		0.08	−2.59, 2.74
30–119 min/d	147 (2.36)	0.17	0.02, 1.22	0.58	−0.64, 1.79
120–239 min/d	178 (2.86)	0.57	0.20, 1.60	−0.93	−2.05, 0.20
≥240 min/d	427 (6.87)	0.31	0.13, 0.79 *	−0.85	−1.64, −0.05 *
MPA					
0 min/d	4796 (77.16)	1		1	
10–29 min/d	199 (3.20)	0.31	0.11, 0.84 *	2.49	1.41, 3.56 **
30–119 min/d	551 (8.86)	0.21	0.09, 0.45 **	2.57	1.85, 3.29 **
120–239 min/d	332 (5.34)	0.51	0.25, 1.05	1.52	0.63, 2.41 **
≥240 min/d	338 (5.44)	0.12	0.03, 0.48 **	1.87	0.97, 2.76 **
LPA					
0 min/d	331 (5.32)	1		1	
10–29 min/d	1088 (17.50)	0.75	0.44, 1.27	1.45	0.62, 2.29 **
30–119 min/d	496 (7.98)	0.71	0.50, 1.02	1.20	0.64, 1.76 **
120–239 min/d	3994 (64.25)	0.76	0.47, 1.23	0.49	−0.25, 1.22
≥240 min/d	307 (4.94)	0.64	0.31, 1.36	0.52	−0.41, 1.45
Volume					
VPA					
0 min/w	5446 (87.61)	1		1	
10–74 min/w	11 (0.18)	N/A		−0.81	−5.09, 3.47
75–299 min/w	90 (1.45)	N/A		0.67	−0.85, 2.19
≥300 min/w	669 (10.76)	0.40	0.21, 0.78 **	−0.86	−1.51, −0.21 *
MPA					
0 min/w	4810 (77.38)	1		1	
10–149 min/w	291 (4.68)	0.21	0.08, 0.58 **	2.52	1.61, 3.43 **
150–299 min/w	101 (1.62)	0.56	0.17, 1.82	3.53	2.07, 4.99 **
≥300 min/w	1014 (16.31)	0.25	0.14, 0.44 **	1.91	1.31, 2.51 **
LPA					
0 min/w	4012 (64.54)	1		1	
10–149 min/w	368 (5.92)	0.84	0.51, 1.37	1.58	0.78, 2.38 **
150–299 min/w	94 (1.51)	1.10	0.46, 2.62	2.07	0.58, 3.56 **
≥300 min/w	1742 (28.02)	0.66	0.49, 0.90 **	0.92	0.42, 1.42 **

Abbreviations: OR, odds ratio; β, Regression coefficients; CI, confidence interval; PA, physical activity; VPA, vigorous physical activity; MPA, moderate physical activity; LPA, light physical activity; Model was adjusted for age, sex, educational status, marital status, drinking, smoking, BMI; N/A denoted that no applicable value was observed; *: *p* < 0.05; **: *p* < 0.01.

**Table 3 ijerph-18-12698-t003:** Associations between Mets and the daily physical activity ability and cognitive function of patients with heart disease.

Variables	*n* (%)	Daily Physical Function		Cognitive Function	
		OR	95% CI	β	95% CI
METs					
0–599	3985 (64.11)	1		1	
600–1199	99 (1.59)	0.57	0.25, 1.32	2.79	1.35, 4.23 **
1200–1799	397 (6.39)	0.64	0.42, 0.96 *	1.48	0.73, 2.22 **
1800–2999	200 (3.22)	0.10	0.02, 0.39 **	3.28	2.25, 4.31 **
3000–5999	599 (9.64)	0.36	0.24, 0.56 **	1.77	1.15, 2.39 **
6000–8999	308 (4.95)	0.13	0.05, 0.34 **	2.10	1.26, 2.94 **
9000–11,999	157 (2.53)	0.13	0.03, 0.52 **	1.99	0.84, 3.14 **
≥12,000	471 (7.58)	0.15	0.07, 0.31 **	1.42	0.73, 2.12 **

Abbreviations: OR, odds ratio; β, Regression coefficients; CI, confidence interval; Model was adjusted for age, sex, educational status, marital status, drinking, smoking, and BMI; *: *p* < 0.05; **: *p* < 0.01.

## Data Availability

The dataset collected and analyzed in the current study are available from the corresponding author on reasonable request.

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
