# Peer review of "The Dose–Response Relationships of Different Dimensions of Physical Activity with Daily Physical Function and Cognitive Function in Chinese Adults with Hypertension: A Cross-Sectional Study"

_ijerph, 2021, doi:10.3390/ijerph182312698_

Round 1
Reviewer 1 Report
no suggestions. the review is OK.
Author Response
Thank you for your review of the manuscript.
Reviewer 2 Report
The physical activity guidelines for the general population as well as hypertensive patients have been already established from a serious of robustly designed longitudinal studies over the years.
Hypertensive patients may need to undergo an initial screening procedure in order to recommend the level of physical activity that best suits them, if at all.
Besides, vigorous physical activity appears to do more harm than good to hypertensive patients contrary to what appears to be recommended in this study.
There is no much new evidence to make of in this cross-sectional study that would contribute to overruling, changing, or even modifying the established and recommended physical Activity guidelines for hypertensive patients derived from years of more robustly designed longitudinal studies.
References
Sharman, J. E., La Gerche, A., & Coombes, J. S. (2015). Exercise and cardiovascular risk in patients with hypertension. American journal of hypertension, 28(2), 147-158. Accessed at https://academic.oup.com/ajh/article/28/2/147/2730195
Zaleski, A.( 2019, February 7). Exercise for the Prevention and Treatment of Hypertension - Implications and Application. American College of Sports Medicine. Accessed at
https://www.acsm.org/blog-detail/acsm-certified-blog/2019/02/27/exercise-hypertension-prevention-treatment
Author Response
Thank you for your valuable suggestions. We quite agree with your point of view. Our results are close to the current hypertension exercise guidelines. The current exercise guidelines pay more attention to hypertension related cardiovascular and cerebrovascular diseases, however, our research focused more on the effects of exercise on the daily physical function (DPF) and cognitive function (CF) of patients with hypertension.
Reviewer 3 Report
Abstract: it should be reformulated. Remove all those formulas, that distract the reader and do not allow a flow reading. Leave only the most significant formulas and describe the results in a more fluent way.
Introduction: when abbreviations are mentioned in the text for the first time they should be written in full, otherwise the reading becomes too complex, e.g., lines 61, 92, etc.
References: half of them are too dated, so they should be revised and modified.
Conclusions: they should not be a repetition of the results, so it is not necessary to cite numbers, but to make final considerations on the quality of the results and the impact of the proposed study.
Abbreviations: add a list of all the abbreviations mentioned in the text at the bottom before the references.
Author Response
Thank you for your comments and suggestions. We have made changes one by one according to your opinion.
Q1, Abstract: it should be reformulated. Remove all those formulas, that distract the reader and do not allow a flow reading. Leave only the most significant formulas and describe the results in a more fluent way.
Answer: we have removed the formulas in abstract part.
Q2, Introduction: when abbreviations are mentioned in the text for the first time they should be written in full, otherwise the reading becomes too complex, e.g., lines 61, 92, etc.
Answer: we have revised the first appearance abbreviations in the text. Please see lines 46,54,154.
Q3, References: half of them are too dated, so they should be revised and modified.
Answer: we have renewed the references.
Q4, Conclusions: they should not be a repetition of the results, so it is not necessary to cite numbers, but to make final considerations on the quality of the results and the impact of the proposed study.
Answer: we have revised the conclusions, please see line 335-340.
Q5, Abbreviations: add a list of all the abbreviations mentioned in the text at the bottom before the references.
Answer: we have added a list of all the abbreviations in the end of text, please see line 341-345.
This manuscript is a resubmission of an earlier submission. The following is a list of the peer review reports and author responses from that submission.
Round 1
Reviewer 1 Report
- Thank you for working on an important issue to address the dose-response relationship of PA. Should not the title be ............different dimensions of PA with DPD and CH?
- The major issue is that the odds ratio (DPF) and Beta coefficients (CF) are presented in the same Table for two different outcome variables. I would suggest presenting the odds ratio for both outcome variables to make a meaningful comparison. In the abstracts, Beta coefficients, odds ratio, different outcomes, and different levels of exposure have made it difficult to follow. Please simplify.
- section 2.2.1 screening of hypertensive patients needs to expand. Curretnly, it is not very clear. Who are the eligible patients? Who said they were diagnosed and on medication. Does it mean if patients are not on medication, they are not eligible. Please elaborate.
- Line 153: BMI was 25.26 years.
- Which questionnaire was used? Is it WHO GPAQ? The following article could be consulted to compare some findings.
Islam F.M.A: Factors Associated with Physical Activity among People with Hypertension in a Rural Area in Bangladesh: Baseline Data from a Cluster Randomized Control Trial. International journal of environmental research and public health 2021, 18(14).
- Tables 2 & 3. I would suggest using the odds ratio for both outcomes
- The dose-response does not show a clear trend. For example, Table 2, LPA. 1.34, 2.01, 0.94. what is the trend? These are true for many other cases. PLEASE insert the P values for trends where there are significant dose responses.
- Please insert a column and present the number and percentage in both Tables 2 and 3. For example, 0d/w = 200 (5%), 1-2d/w = 400 (10%)......and so on.
Author Response
Dear Reviewer,
Thank you very much for your comments concerning our manuscript. Those comments are all valuable and very helpful for revising and improving our paper, as well as the important guiding significance to our researches. We have studied comments carefully and have made correction which we hope meet with approval. Revised portion are marked in red in the paper. The main corrections in the paper and the responds to your comments are as flowing:
- Thank you for working on an important issue to address the dose-response relationship of PA. Should not the title be ............different dimensions of PA with DPD and CH?
*Answer: Thanks for your good suggestions. We have revised in the manuscript, please see the title. Thank you.
- The major issue is that the odds ratio (DPF) and Beta coefficients (CF) are presented in the same Table for two different outcome variables. I would suggest presenting the odds ratio for both outcome variables to make a meaningful comparison. In the abstracts, Beta coefficients, odds ratio, different outcomes, and different levels of exposure have made it difficult to follow. Please simplify.
*Answer: Thanks for your good suggestions. The cognitive function we studied was a continuous variable. We used linear regression. There was no report of odds ratio value in this statistical method. Daily physical function is a categorical variable, and we use logistic regression, so it is reasonable to report odds ratio. In addition, although we use odds ratio and beta coefficients in one table, they are the results of different statistical analysis for different dependent variables. In addition, different dependent variables cannot be compared, but can only be compared in a single dependent variable. Therefore, we believe that this is feasible.
- section 2.2.1 screening of hypertensive patients needs to expand. Curretnly, it is not very clear. Who are the eligible patients? Who said they were diagnosed and on medication. Does it mean if patients are not on medication, they are not eligible. Please elaborate.
*Answer: We are deeply sorry about the unclear description. We have revised in the manuscript, please see section 2.2.1. Thank you.
- Line 153: BMI was 25.26 years.
*Answer: We are deeply sorry about this. We have deleted “years” in the manuscript, please see Line 162. Thank you.
- Which questionnaire was used? Is it WHO GPAQ? The following article could be consulted to compare some findings.
Islam F.M.A: Factors Associated with Physical Activity among People with Hypertension in a Rural Area in Bangladesh: Baseline Data from a Cluster Randomized Control Trial. International journal of environmental research and public health 2021, 18(14).
*Answer: Thanks for your good suggestions. The physical activity questionnaire we used is not a mature questionnaire before, and there may be a problem that the measurement is not accurate enough. We have revised in the discussion of manuscript, please see Line 330-332. In addition, there have been published articles using this questionnaire to measure physical activity before [1], so we think it is also feasible. Thank you.
- Yang D, Bian Y, Zeng Z, Cui Y, Wang Y, Yu C. Associations between Intensity, Frequency, Duration, and Volume of Physical Activity and the Risk of Stroke in Middle- and Older-Aged Chinese People: A Cross-Sectional Study. Int J Environ Res Public Health. 2020 Nov 20;17(22):8628. doi: 10.3390/ijerph17228628. PMID: 33233679; PMCID: PMC7699744.
- Tables 2 & 3. I would suggest using the odds ratio for both outcomes
*Answer: Thanks for your good suggestions. Considering that cognitive function variables are continuous variables, we conducted statistical analysis through linear regression, which has no odds ratio, therefore, we report aegression coefficients.
- The dose-response does not show a clear trend. For example, Table 2, LPA. 1.34, 2.01, 0.94. what is the trend? These are true for many other cases. PLEASE insert the P values for trends where there are significant dose responses.
*Answer: Thanks for your good suggestions. The non standardized coefficient in linear regression indicates how many units the dependent variable will increase / decrease for each unit of an independent variable. However, if the independent variable is not statistically significant enough, there is no difference between this coefficient and 0. Whether it is significant or not, we refer to its corresponding t value and P value. For example, the math coefficient (parameter estimation) is. 389. This means that for every 1 point increase in math score, it can be estimated that the science score will increase by. 389 points, or. 389 units, provided that all other independent variables remain unchanged. Therefore, it is also feasible to look at the non-standard coefficients in the linear regression results.
- Please insert a column and present the number and percentage in both Tables 2 and 3. For example, 0d/w = 200 (5%), 1-2d/w = 400 (10%)......and so on.
*Answer: Thanks for your good suggestions. We have inserted a column and presented the number and percentage in both Tables 2 and 3, please see Tables 2 and 3. Thank you.

Reviewer 2 Report
This reviewer commends the authors for this interesting study:
- How were the participants screened for morbidities that affect baseline cognitive function such as early-onset Alzheimer’s Disease and other dementias?
- How were the participants screened for morbidities that restrict physical activity such as degenerative diseases of the joints?
- The subjective classification and reporting of physical activity by participants into Vigorous Physical activity (VPA), moderate physical activity (MPA), and Light Physical Activity (LPA) is purely speculative. What appears to be a moderate physical activity by one physically more fit study participant could be labeled a vigorous physical activity by another physically less fit (frailer) study participant. How do the researchers account for these plausible discrepancies?
- The significance of the bias introduced in the design and method of the research is evident in the type of responses that could be gathered from study participants, which in turn is reflected in the results of the study. For example, on page 8, line 188-191, the results indicate that “Compared 188 with 0 min/week, patients with more than 300 min/week of vigorous physical activity (β=- 189 0.86,CI:-1.51,-0.21) had lower CF scores, and patients with moderate and LPA at any 190 volume had higher CF scores (P <0.05).” (emphasis mine). How do the authors account for the fact that participants who are physically active have lower Cognitive Function Scores than sedentary participants of this study?
- What is/are the research question (s) and/or what hypothesis/hypotheses are the researchers trying to address in this study?
Author Response
Dear Reviewer,
Thank you very much for your comments concerning our manuscript. Those comments are all valuable and very helpful for revising and improving our paper, as well as the important guiding significance to our researches. We have studied comments carefully and have made correction which we hope meet with approval. Revised portion are marked in red in the paper. The main corrections in the paper and the responds to your comments are as flowing:
This reviewer commends the authors for this interesting study:
- How were the participants screened for morbidities that affect baseline cognitive function such as early-onset Alzheimer’s Disease and other dementias?
*Answer: The cognitive function in this paper is measured by a mature scale, and the purpose is to explore the dose-response relationship between different dimensions of physical activity and cognitive function. Therefore, considering that some diseases affecting cognitive function are not the purpose of this study, we did not screen diseases affecting baseline, such as dementia. This is the limitation of this study. We have revised in the manuscript. Please see line 332-334.
- How were the participants screened for morbidities that restrict physical activity such as degenerative diseases of the joints?
*Answer: The purpose of this study is to explore the dose-response relationship between different dimensions of physical activity including intensity, time, frequency, capacity and metabolic equivalent and DPF and CF in chinese adults with hypertension. Limiting physical activity diseases is not the main purpose of this study. Therefore, we did not conduct screening, but considering that it may have a certain impact on the results of this study, we have revised in the discussion part, please see line 332-334.
- The subjective classification and reporting of physical activity by participants into Vigorous Physical activity (VPA), moderate physical activity (MPA), and Light Physical Activity (LPA) is purely speculative. What appears to be a moderate physical activity by one physically more fit study participant could be labeled a vigorous physical activity by another physically less fit (frailer) study participant. How do the researchers account for these plausible discrepancies?
*Answer: Thanks for your good suggestions. Exercise intensity refers to the. amount of force and the tension of the body. We define different intensities of exercise according to the different feelings of each person during exercise. During the measurement, the physical feeling of each exercise intensity is described. For example, high-intensity exercise will make you breathe fast, such as carrying heavy objects, digging, farming, aerobic exercise, fast cycling, cycling and loading goods. Medium intensity exercise makes you breathe faster than usual, such as carrying light things, cycling at conventional speed, mopping the floor, playing Taijiquan Gallop, etc. Therefore, constant exercise intensity must be different exercise intensity for people in different body states. This is also what we call personalized exercise prescription. It varies from person to person. The same exercise time and frequency will make different people have different physical reactions. However, the subjective questionnaire measurement is not accurate enough, and we have supplemented the defects in people.
- The significance of the bias introduced in the design and method of the research is evident in the type of responses that could be gathered from study participants, which in turn is reflected in the results of the study. For example, on page 8, line 188-191, the results indicate that “Compared 188 with 0 min/week, patients with more than 300 min/week of vigorous physical activity (β=- 189 0.86,CI:-1.51,-0.21) had lower CF scores, and patients with moderate and LPA at any 190 volume had higher CF scores (P <0.05).” (emphasis mine). How do the authors account for the fact that participants who are physically active have lower Cognitive Function Scores than sedentary participants of this study?
*Answer: We are sorry about this unclear description. This study found that. patients with high blood pressure who had a frequency of strenuous physical activity of 6 to 7 days per week, a duration of at least 240 minutes per day, and a volume of at least 300 minutes per week were less likely to have impaired DPF, but poorer CF. Therefore, VPA should be recommended carefully in Chinese adults with hypertension. please see line 340-344. In addition, we explained the reasons for this result in the discussion section, please see line 232-243.
- What is/are the research question (s) and/or what hypothesis/hypotheses are the researchers trying to address in this study?
*Answer: Thanks for your good suggestions. We have added “We hypothesized. that there was a dose–response relationship between different di-mensions of physical activity (including intensity, time, frequency, volume and metabolic equivalent) and DPF and CF in Chinese adult patients with hypertension.” in the manuscript, please see line 61-64. Thank you.

Reviewer 3 Report
Thank you for this interesting paper. What an interesting data set. I wondered if you are able to do longitudinal research on this topic with this data set.
You can improve the readability of the paper by making the corrections suggested in the attached PDF.

Author Response
Dear Reviewer,
Thank you very much for your comments concerning our manuscript. Those comments are all valuable and very helpful for revising and improving our paper, as well as the important guiding significance to our researches. We have studied comments carefully and have made correction which we hope meet with approval. Revised portion are marked in red in the paper. The main corrections in the paper and the responds to your comments are as flowing:
- Thank you for this interesting paper. What an interesting data set. I wondered if you are able to do longitudinal research on this topic with this data set.
*Answer: Thanks for your good suggestions. We will continue to conduct longitudinal research on this topic with this data set in the future. Thank you.
- You can improve the readability of the paper by making the corrections suggested in the attached PDF.
*Answer: Thanks for your good suggestions. We have revised in the manuscript, please see line 101; line 105-106; line 124; line 130-132; line 139; line 142; line 163-165; line 219; line 221; line 230; line 242-247; line 251; line 330-336; line 351. Thank you.

Reviewer 4 Report
This research forms a solid cohort, and various research results can be expected. However, I have decided that this study cannot be published for the following reasons.
- It is hard to say that the reason and rationality for focusing on the population with hypertension in this study is clear. In addition, although it seems that they have diseases other than hypertension, it cannot be said that the purpose of this study has been achieved because no studies have been made in the analysis of this study.
- The title says "older adults". Since the original cohort is formed over the age of 45, the blood pressure competition population is only older than the target for "older adults", so it is doubtful how to think about the population. Remain.
- This study is a cross-sectional study and no causal relationship can be mentioned. However, in multiple places (for example, lines 27-28, 218-220, 332-334, etc.), the sentences are inappropriate because they assume a causal relationship.
Author Response
Dear Reviewer,
Thank you very much for your comments concerning our manuscript. Those comments are all valuable and very helpful for revising and improving our paper, as well as the important guiding significance to our researches. We have studied comments carefully and have made correction which we hope meet with approval. Revised portion are marked in red in the paper. The main corrections in the paper and the responds to your comments are as flowing:
- It is hard to say that the reason and rationality for focusing on the population with hypertension in this study is clear. In addition, although it seems that they have diseases other than hypertension, it cannot be said that the purpose of this study has been achieved because no studies have been made in the analysis of this study.
*Answer: Thanks for your good suggestions. Firstly, in the introduction, we introduce the reasons why this study focuses on hypertension, and describe the screening process of patients with hypertension later. In addition, we added “we did not consider other diseases (such as early-onset Alzheimer’s disease and other dementias and degenerative diseases of the joints), which may have had an impact on our results.”in the discussion of limitations section, please see line 332-334. Thank you.
- The title says "older adults". Since the original cohort is formed over the age of 45, the blood pressure competition population is only older than the target for "older adults", so it is doubtful how to think about the population. Remain.
*Answer: Thanks for your good suggestions. We have revised the "older adults" to “Chines Adults” in the manuscript. Thank you.
- This study is a cross-sectional study and no causal relationship can be mentioned. However, in multiple places (for example, lines 27-28, 218-220, 332-334, etc.), the sentences are inappropriate because they assume a causal relationship.
*Answer: Thanks for your good suggestions. We have made assumptions in the introduction. Therefore, we have made such a description later and similar articles have been published [1,2]. Therefore, we think it is feasible.
- Dong X, Ding M, Chen W, Liu Z, Yi X. Relationship between Smoking, Physical Activity, Screen Time, and Quality of Life among Adolescents. Int J Environ Res Public Health. 2020 Oct 31;17(21):8043. doi: 10.3390/ijerph17218043. PMID: 33142847; PMCID: PMC7662320.
- Yi X, Liu Z, Qiao W, Xie X, Yi N, Dong X, Wang B. Clustering effects of health risk behavior on mental health and physical activity in Chinese adolescents. Health Qual Life Outcomes. 2020 Jul 3;18(1):211. doi: 10.1186/s12955-020-01468-z. PMID: 32620107; PMCID: PMC7333302.
